# The Mission Support System (MSS v7.0.4) and its use in planning for the SouthTRAC aircraft campaign

Reimar Bauer[1,3], Jens-Uwe Grooß[1,3], Jörn Ungermann[1,3], May Bär[1,3], Markus Geldenhuys[1,3], and Lars Hoffmann[2,3]

[1]Institut für Energie- und Klimaforschung (IEK-7), Forschungszentrum Jülich GmbH, Jülich, Germany
[2]Jülich Supercomputing Centre, Forschungszentrum Jülich GmbH, Jülich, Germany
[3]Center for Advanced Simulation and Analytics (CASA), Forschungszentrum Jülich GmbH, Jülich, Germany

**Correspondence:** j.ungermann@fz-juelich.de

**Abstract.** The Mission Support System (MSS) is an open source software package that has been used for planning flight tracks of scientific aircraft in multiple measurements campaigns during the last decade. It consists of three major components: a web map server located close to the model data storage site that is capable of producing a variety of 2-D figures from 4-D meteorological data; a client application capable of displaying the figures in combination with the planned flight track and an assortment of additional information; a new collaboration server component that enables real-time collaboration of multiple remote parties. During the last decade, these components were constantly improved towards being simple to setup and use and standard compliant.

Here, we describe the use of MSS during the Southern Hemisphere Transport, Dynamics, and Chemistry–Gravity Waves (SouthTRAC-GW) measurement campaign in 2019. This campaign, based in Rio Grande, Argentina, used the German research aircraft HALO to investigate several scientific objectives related to the Southern Hemisphere chemistry and dynamics. We present the diverse data products offered by the MSS web map server dedicated to the campaign, which were derived from European Centre for Medium-Range Weather Forecasts (ECMWF) forecast data, Chemical Lagrangian Model of the Stratosphere (CLaMS) simulations, and Atmospheric Infrared Sounder (AIRS) near-realtime brightness temperature measurements. As example for how the MSS software is used in conjunction with the different data sets, we describe the planning of a single flight, which eventually took place on the 12th of September 2019, probing orographic gravity waves propagating up into the lower mesosphere.

## 1   Introduction

Planning and executing aircraft-based scientific measurement campaigns is an involved process. These campaigns take years to prepare and operate typically with a very constrained budget on both execution time and money. The more important it is to exploit each possible flight to the fullest. This requires thorough planning and leveraging forecast data by operational and scientific atmospheric models. Typically, scientific research flights are conducted to answer a set of defined scientific questions. Predictions of relevant parameters by a model simulation for a particular location provide guidance on where to fly to answer these scientific questions through measurements. Many other constraints concerning, e.g., flight altitude and range, ambient

temperature, overflight permits, secondary airports, down to penguin protection zones, or commercial flight corridors need be observed.

One tool that simplifies the task of planning scientific measurements flights is the Mission Support System (MSS v7.0.4; Bauer et al., 2022). It includes powerful features to allow scientists to plan measurement flights remotely and in the field based on the characteristics of their instruments and model data. In 2010, the Mission Support System was developed by Marc Rautenhaus at Deutsches Zentrum für Luft- und Raumfahrt (DLR) in collaboration with Forschungszentrum Jülich (FZJ) (Rautenhaus et al., 2012). Since then, MSS has become open source. The software essentially consists of three major components: a graphical user interface (GUI) to display data and plan the flights; a server to provide maps and cross-sectional plots through model data; a server to facilitate real-time collaboration from multiple sites.

The software was employed in many campaigns during the last decade, the major ones being: the Midlatitude Cirrus experiment (ML-Cirrus; e.g. Voigt et al., 2017) in 2014, the Polar Stratosphere in a Changing Climate campaign in 2015 (POL-STRACC; Oelhaf et al., 2019), North Atlantic Waveguide and Downstream Impact Experiment (NAWDEX; e.g. Schäfler et al., 2018) in 2016, the Stratospheric and upper tropospheric processes for better Climate predictions campaign in 2017 (Strato-Clim; e.g. Höpfner et al., 2019) the Wave driven ISentropic Exchange campaign in 2017 (WISE; e.g. Kunkel et al., 2019), the Carbon Dioxide and Methane Mission (CoMet 1.0; e.g. Fix et al., 2018) campaign in 2018, the Effect of Megacities on the transport and transformation of pollutants on the Regional and Global scales in 2018 (EMeRGe; Andrés Hernández et al., 2022), the Southern Hemisphere Transport, Dynamics, and Chemistry–Gravity Waves campaign in 2019 (SouthTRAC; Rapp et al., 2021), the CIRRUS in High Latitudes (CIRRUS-HL) in 2021, the Arctic Air Mass Transformations During Warm Air Intrusions And Marine Cold Air Outbreaks (HALO-(AC)[3]) campaign in 2022, and so far finally the Development and Testing of Airborne Laminar Flow Inlet for Condensable Vapors in 2022 (TI3GER).

This paper describes the recent development of this software in Sec. 2 and, as an example for its use in practice, its use in planning flights for the SouthTRAC campaign in Sec. 3. A summary is given in Sec. 4.

## 2 MSS flight planning tool suite

The MSS flight planning tool suite is an open source software package designed to combine a powerful flight path designer with a host of overlay features to combine horizontal and vertical cross-sections through model data along the flight path. In particular, it is designed to allow remote operation, i.e., the planning can take place at a location that is far away from the location of the typically bulky forecast model data and therefore avoids excessive data transfers between the sites of the campaign and the data centre.

It does so by providing three major components:

1. a server component (Mission Support Web Map Server; MSWMS) to deliver images of horizontal and vertical cross-sections through model data. This builds on top of an Open Geospatial Consortium (OGC) Web Map Service standard compliant server (for Standardisation, 2005), enhanced by MSS-specific features allowing for vertical cross sections or sampling 4-D data along the flight path.

2. a real-time collaboration server (Mission Support Collaboration server; MSColab), which allows multiple clients to simultaneously work on a single flight plan stored on a server.

3. a client component (Mission Support User Interface; MSUI) implementing a OGC Web Map Service client (which is
also able to connect to third-party servers) in combination with a visual flight planning tool. This is supplemented by several plugins dedicated to a variety of typical tasks of scientific flight planning, such as, e.g., visualising solar angles, or computing the effect of dive maneuvers on the range of the aircraft.

The software project also contains several ancillary components, such as an exemplifying set of scripts describing how to automatically download and process weather forecast data from European Centre for Medium-Range Weather Forecast (ECMWF)
for use in the server component or a plugin to the Windy website (Windy, 2022) connecting to the real-time collaboration server.

## 2.1 Architecture

Figure 1 shows the interaction of the main components MSUI, MSWMS, and MSColab.

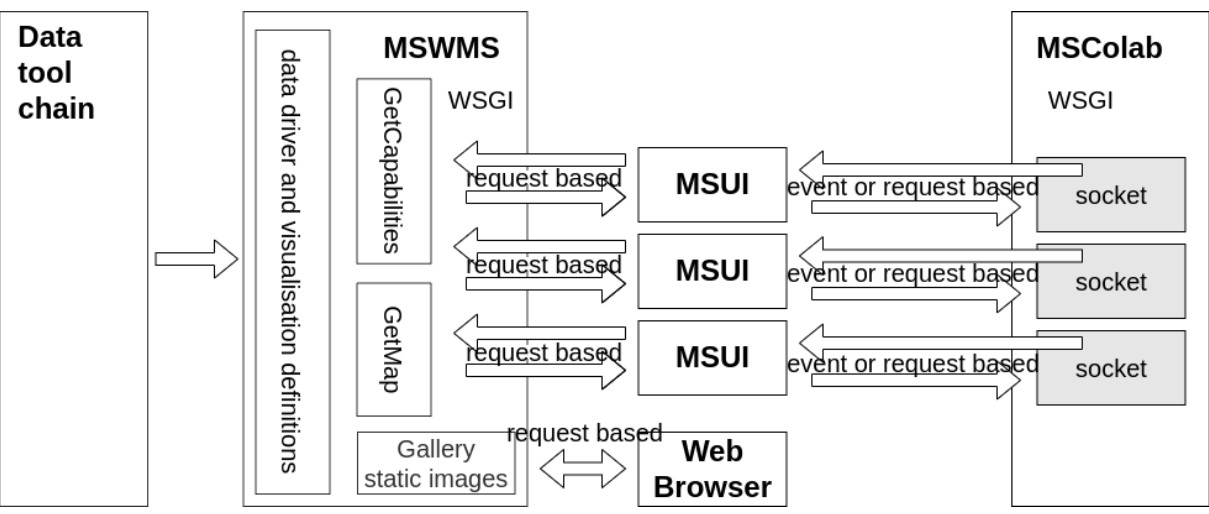

**Figure 1.** Data flow and architecture of the Mission Support System.

The client application allows the users to interact with the servers. On one side, it allows to retrieve arbitrary images via
the http protocol from the MSWMS component. On the other side, it communicates with the MSColab server to modify flight plans, use the chat, etc. A detailed description of the individual components follows.

## 2.2 Mission Support Web Map Server (MSWMS)

The MSWMS data server implementation follows the OGC Web Map Service (OGC WMS) standard (for Standardisation, 2005). This is a standard for geoscientific data formats combined with a web service for generating images of maps. It specifies,

e.g., what kind of projections must be supported by the server. Using it allows us to generate graphics based on defined HTTP requests from meteorological data files in a standardized fashion. Since OGC WMS is an open standard, there are a large number of third party servers available that offer, e.g., satellite imagery that has also proven useful for flight planning in the past.

    The MSWMS server is designed to be updated with new data from model calculations on a regular basis, so that the user is

offered the most recent data as soon as it becomes available.

    The standard prescribes two main functions: with the *GetCapabilities* request, the user receives a catalog of available products in the form of an Extensible Markup Language (XML) file. The *GetMap* request contains parameters such as the map section and projection, the desired product/layer, the time, etc.; the server replies with a graphic in the compressed Portable Network Graphics format generated according to the specification. In this way, the amount of transferred data is limited to the

size of an compressed image, which is negligible in comparison to the 4-D model data available to the server.

    The OGC WMS standard only defines horizontal maps. But our MSWMS server can also produce vertical cross-sections following a series of longitude/latitude coordinates, i.e., a flight path. This is a key feature of the software that sets it apart from campaign supporting web pages or galleries providing standardized horizontal maps for a predefined region. This feature greatly helps in defining desired flight levels or predict the outcome of, e.g., vertical lidar measurements. A new feature is also

the sampling of model data along a one-dimensional flight path, which allows for correlation plots in the client or predicting the measurements of in situ instruments.

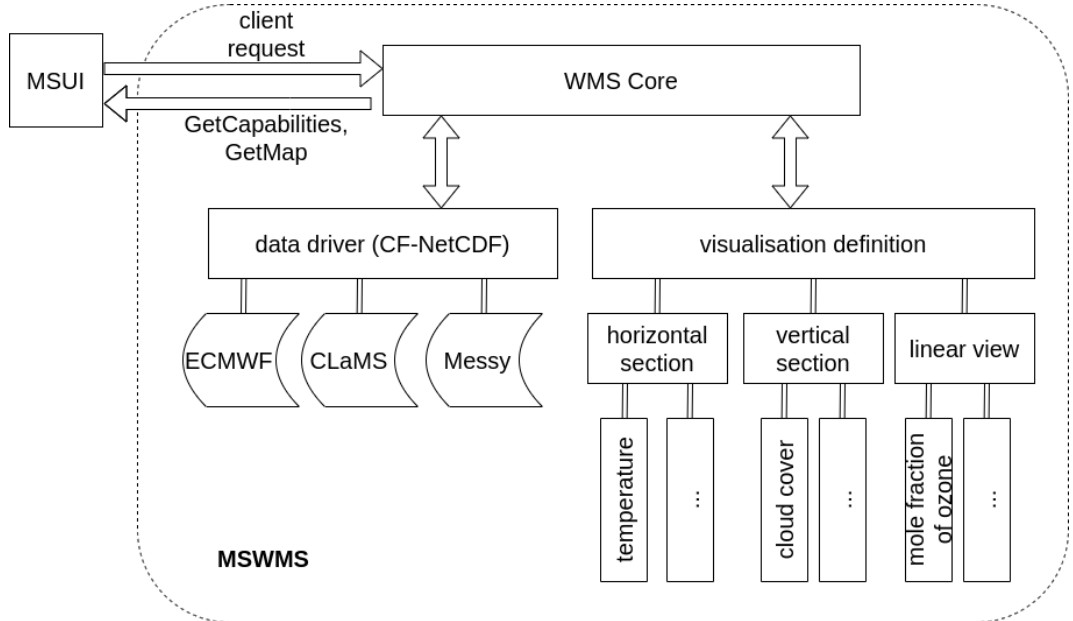

**Figure 2.** Data driver and visualisation definitions for the MSWMS Server

The architecture of the server component is shown in Fig. 2. The left hand side show a generic data processing tool chain that supplies geophysical data to the server. A simple tool chain is provided as an example to show how to provide MSS-compliant model data from ECMWF Reanalysis v5 (ERA5) and operational data (Hersbach et al., 2020; Mission Support

System (MSS) data retrieval on github). The *WMS core* component is implemented as Flask Web Server Gateway Interface (WSGI) component for use in any WSGI Web server.

This handles mostly the two request types described above. The server is fully configurable and modifieable by a configuration Python script. This script configures, amongst other things, the desired plotting layers and the location of data stored in the self-describing Network Common Data Form (NetCDF). The plotting layers are a set of Python classes that perform the

100 actual task of cross-section plotting of geophysical quantities. The *data driver* component parses all data found in specified directories and extracts provided data fields leveraging the Climate and Forecast metadata conventions (Hassell et al., 2017). Plotting layers of the *visualisation definition* component are connected with needed data using the *standard_name* attribute observing the units specified in the *units* attribute of the NetCDF file. It is possible and very simple to define new plotting layers from within the server configuration file.

In addition to the OGC WMS service, the MSWMS component also provides a configurable web gallery of the provided layers and data, which can be used without installing the client component (e.g., from a smart phone). While being of practical use, this feature also allows the server maintainer to quickly plot all configured layers for testing purposes.

### 2.3 Mission Support User Interface (MSUI)

The MSUI component supports the planning of flight tracks along a series of geospatial coordinates.

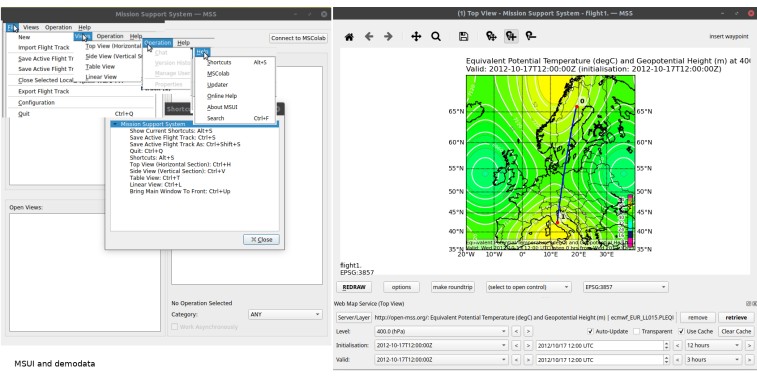

**Figure 3.** Main window of the MSUI.

The main interface is shown in Fig. 3. On the left hand is the flight track management and MSColab interface. On the right hand side is a *top view* window with a simple flight path consisting of only two waypoints on top of a visualisation of test data. The waypoints can be interactively modified to align with features on the displayed images from the MSWMS component. The vertical movement of the aircraft can be defined in the *side view*. The MSWMS component provides vertical cross-sections to

this purpose. The third currently supported *linear view* visualizes model data along the flight path similar to the measurements an in-situ instrument would make. Section 3.1 provides ample examples for all types of plots.

All views can be opened simultaneously and multiple times to display different information on each. The user interface of the individual views has an interface for selecting different OGC WMS servers. This allows the combination of different and multiple maps/layers from one or several OGC WMS servers into one view (see Sec. 3.2 for an example). In addition to the images provided by OGC WMS servers, further information can be displayed by a series of widgets, e.g., Keyhole Markup Language (KML; Consortium, 2015) data, airports/airdata, or satellite tracks.

User specific configuration like favourite OGC WMS servers, predefined map sections, etc. can be stored permanently in a local configuration file. Typically, a default configuration file containing the relevant servers and access details for a campaign is prepared and distributed to all users for initialising their flight planning environment.

| | Location | Lat (+-90) | Lon (+-180) | Flightlevel | Pressure (hPa) | Leg dist. (km [nm]) | Cum. dist. (km [nm]) | Leg time | Cum. time | Time (UTC) | Rem. fuel (lb) | Aircraft weight |
|---|---|---|---|---|---|---|---|---|---|---|---|---|
| 0 | Rio Grande | -53.78 | -67.7 | 0 | 1,013.25 | 0 [0] | 0 [0] | 00:00:00 | 00:00:00 | 2022-08-22 19:21:34 | 35000 | 91000 |
| 1 | | -53.78 | -67.7 | 350 | 238.42 | 0 [0] | 0 [0] | 00:01:30 | 00:01:30 | 2022-08-22 19:23:05 | 34024 | 90024 |
| 2 | | -47.96 | -77.35 | 350 | 238.42 | 937 [506] | 937 [506] | 01:08:50 | 01:10:21 | 2022-08-22 20:31:55 | 30164 | 86164 |
| 3 | | -47.96 | -77.35 | 410 | 178.74 | 0 [0] | 937 [506] | 00:00:07 | 01:10:28 | 2022-08-22 20:32:03 | 30045 | 86045 |
| 4 | | -48.67 | -76.34 | 410 | 178.74 | 108 [58] | 1045 [564] | 00:07:40 | 01:18:08 | 2022-08-22 20:39:43 | 29637 | 85637 |
| 5 | | -51.66 | -71.64 | 410 | 178.74 | 472 [255] | 1518 [819] | 00:33:19 | 01:51:28 | 2022-08-22 21:13:03 | 27874 | 83874 |
| 6 | | -55.63 | -63.82 | 410 | 178.74 | 679 [366] | 2197 [1186] | 00:48:03 | 02:39:31 | 2022-08-22 22:01:06 | 25381 | 81381 |
| 7 | | -59.84 | -51.18 | 410 | 178.74 | 884 [477] | 3082 [1664] | 01:02:51 | 03:42:22 | 2022-08-22 23:03:57 | 22208 | 78208 |
| 8 | | -54.77 | -52.66 | 410 | 178.74 | 571 [308] | 3654 [1973] | 00:40:51 | 04:23:14 | 2022-08-22 23:44:49 | 20220 | 76220 |
| 9 | | -54.77 | -52.66 | 470 | 133.96 | 0 [0] | 3654 [1973] | 00:00:33 | 04:23:47 | 2022-08-22 23:45:22 | 20124 | 76124 |

(select to open control) ▾    Waypoints:   insert   clone   delete selected   reverse   make roundtrip

**Figure 4.** The table view window of the MSUI.

In the *table view*, which resembles a spreadsheet, all waypoints are displayed numerically. Here, exact coordinates can be entered. When performance data of the aircraft is provided, also estimates of range and peak altitude of the aircraft can be computed and displayed in the table. Figure 4 shows an example for the flight discussed in Sec. 3.2. The first column allows to select pre-defined named waypoints or NAVIDS, or to name a waypoint for the current session. The next columns allow to specify longitude, latitude, and flight level exactly. A performance file for the employed aircraft was loaded that allows for the calculation of a rough approximation of flight time and remaining fuel, i.e. range.

The resulting flight track can be exported or imported by different formats. MSUI supports text, comma-separated values (CSV), and KML but other formats can be added in a simple fashion by providing plugins in the configuration directory (with examples being provided in the documentation).

## 2.4 Mission Support Collaboration tool (MSColab)

A recent improvement of MSS is the remote collaboration server for sharing all flight planning data in a synchronized usage. This feature has been put to practical use during the CIRRUS in High Latitudes (CIRRUS-HL) campaign in 2021, which was heavily affected by the Corona pandemic, posing a need for virtual rather than in-person interactions for campaign planning.

The server is based on Flask-SocketIO and can be deployed for example by a Gunicorn Web Server with an eventlet worker. Usually NGINX is then used as a WebSocket Reverse Proxy. The MSColab server is configured by a Python script, very similar to the MSWMS server. The user can specify here the database driver and, for example, the Cross-Origin Resource Sharing (CORS) origins for communications with your server.

The MSColab component allows multiple users to jointly work on a single *operation*. An operation is thereby the general activity of performing a measurement flight including in particular its flight track, but extends this with additional features: Each operation consists of a (version) history of the contained flight track, a fine grained user permission mode, and an associated real-time chat component, which is also used to communicate changes to the managed flight track. Each operation can be assigned different users with different roles: Admin, Creator, Collaborator, and Viewer. E.g., a user with Admin or Creator role can add Users to the operation. A user with Collaborator role can edit the flight path and one with a Viewer Role can only view.

The flight track of the active operation can be simultaneously displayed within the MSUI client for all participants. Depending on their role, each participant may or may not modify it. Each change results in a new revision and all attached MSUI clients are updated immediately. The CORS implementation of the server allows the development of plugins with access to the flight track in other software products, e.g., a prototype plugin for the Windy website Windy (2022) is available.

This feature is fully integrated into the MSUI client, which remains fully functional for designing a flight track also without an MSColab server. But with such a server, MSUI supports multi-user collaboration on operations that are stored on the server.

## 3  Flight planning for the SouthTRAC campaign

The Southern Hemisphere Transport, Dynamics, and Chemistry–Gravity Waves campaign (SouthTRAC-GW; Rapp et al., 2021) took place from September to November 2019 operating from bases located at Oberpfaffenhofen, Germany, and Rio Grande, Argentina. The scientific objectives were very diverse, ranging from examining air polluted by biomass burning in the tropics (Johansson et al., 2022) (and, as target of opportunity, Australia (Ohneiser et al., 2022)), the breakdown of the Antarctic polar vortex and the associated chemical and physical structure, the general chemical composition of the Southern-Hemisphere Upper Troposphere-Lower Stratosphere (Johansson et al., 2022), to studying gravity waves. Some exemplary outcomes from the gravity wave part of the SouthTRAC campaign include: clear-air turbulence studies (Rodriguez Imazio et al., 2022; Dörnbrack et al., 2022), gravity wave propagation from the Southern Andes and the Antarctic peninsula into the mesosphere (Reichert et al., 2021; Conte et al., 2022), gravity wave observation and model intercomparison (Gisinger et al., 2022; Dörnbrack et al., 2022; Liu et al., 2022; Alexander et al., 2022), mountain waves drifting upwind of the Andes (Krasauskas et al., 2022), gravity wave refraction due to wind shear (Geldenhuys et al., 2022) and gravity waves from orographic and non-orographic sources (Alexander et al., 2022; de la Torre et al., 2022).

In this section, we will take a more practical view on the setup of the MSWMS server for the campaign and the use of the MSUI client for the planning of one of the research flights as an example.

## 3.1 Data products

An outstanding part of MSS use is the interplay between scientific mission goals and available data products. The usefulness
of the tool largely depends on defining a comprehensive set of products to be displayed, which must be implemented within the
MSWMS component. Here, we will give a brief overview of the most important mission-specific data products we generated
and/or made available for this campaign.

### 3.1.1 Gravity wave analysis

An important objective of the SouthTRAC campaign was the measurement of gravity wave propagation into the upper at-
mosphere. To that end, we set up a processing chain operating on ECMWF forecast temperature data. First, the background
temperature structure was removed by a detrending routine (Strube et al., 2020). This generates 4-D temperature residuals,
which are largely caused by gravity wave activity. Second, the sine fit 3D (S3D) gravity wave analysis program analysed the
temperature residuals to identify gravity wave parameters such as amplitude, wave length, or momentum flux on three separate
atmospheric layers (Lehmann et al., 2012). Both the residuals and the parameters are offered as layers on the MSWMS server.
Some examples for generated plots are given in Fig. 5.

### 3.1.2 AIRS brightness temperatures

An important part of the post-campaign analysis was envisioned to follow the gravity wave propagation to higher altitudes and
identify the waves measured closer to ground level by aircraft measurements in satellite data. To this end, we implemented
two data products: on the one hand, we added near-real time data of brightness temperature perturbations as measured by the
Atmospheric InfraRed Sounder (AIRS) aboard the National Aeronautics and Space Administration's (NASA) Aqua satellite
(Hoffmann et al., 2013, 2016); on the other hand, we implemented a scheme that predicted brightness temperature perturbations
as detected by AIRS based on the temperature residuals derived from ECMWF forecast data. To that end, we employed Jacobian
matrices computed by the JUelich RApid Spectral SImulation Code (JURASSIC) forward model (Hoffmann and Alexander,
2009) for standardized polar night atmospheric profiles. Multiplying these Jacobian matrices with the temperature residuals
allows to immediately estimate the AIRS brightness temperature perturbations for this situation.

Having both products available allowed us to also examine the reliability of the forecast by comparing the original forecast
perturbations with the actually measured ones. The actual AIRS data also allowed to quickly identify matching structures after
the flights took place to allow a quick first analysis. Some examples are given in Fig. 6. The forecast brightness temperature
residual data product has proven to be largely reliable for forecasting strong gravity wave structures visible to AIRS for the
upcoming days.

### 3.1.3 Further dynamics related plots

Aside the main plots for gravity wave forecasts described above, a number of ancillary plots were provided to be used for analysing the general synoptic situation or for more specific scientific topics. The examples shown below showcase more advanced plotting capabilities of the MSWMS server, which combine multiple data products into one dedicated plot.

The surface weather is depicted in a meteorological standard plot in Fig. 7a, where equivalent potential temperature at 850 hPa is combined with sea level surface pressure contours and geopotential height at 500 hPa. This plots allows to identify frontal systems and estimate their development in the near-future. The second plot in Fig. 7b depicts the state of the Antarctic vortex at 50 hPa, which was used to identify regions, where gravity waves could propagate to high altitudes and to examine the state of the vortex itself during the sudden stratospheric warming event. Panel c of Fig. 7 shows a vertical cross section bringing together several ECMWF data products. The gray colours indicate the static stability in combination with dynamic (light green contours) and thermal (dark green dots) tropopause. The horizontal wind speed (orange contours) allows the identification of jet streams, whereas the presence of clouds (light and dark blue contours) can be used to diagnose radiative effects on the tropopause. This plot is useful for identifying interesting dynamic situations in relation to tropopause inversion layers (see Kunkel et al. (2019) for the inspiration for this plot).

### 3.1.4 CLaMS chemical forecasts

For the SouthTRAC campaign, forecasts of the chemical composition of the atmosphere by the Chemical Lagrangian Model of the Stratosphere (CLaMS; McKenna et al., 2002b, a) have been provided. As a full global integration for each of the envisaged output time steps is numerically rather costly, therefore the forecasts were provided by the Reverse Domain Filling (RDF) method. For that, a regular-spaced latitude/longitude grid for each level in the forecast area is defined for each forecast time step. For those points, Lagrangian back-trajectories are calculated until the time where the global simulation is available. Typically this time is the previous day 12:00 UTC. Only along those trajectories, the CLaMS chemistry scheme is integrated to provide the chemical composition for the above defined grid points in time and space. In addition to the simulation of the chemical composition, also model parameters, like origin tracers and accumulated ozone depletion are provided. Figure 8 shows examples of map projections from the SouthTRAC campaign flight planning. Panel (a) shows the active chlorine $ClO_x$ for the above mentioned gravity wave flight ST08. From this it is clear that for this specific flight, the observation of active chlorine in the polar vortex is not compatible with the other flight goals (see Sec. 3.2). Panels (b) to (d) show flight planning plots for the research flight 25 on 12 November 2019 that was dedicated to probe the structure of a filament of vortex origin and also vortex air itself. Panel (b) shows the vortex origin tracer, a model quantity initialized with 1 inside the vortex and 0 outside and then transported and mixed as the other chemical compounds. Panels (c) and (d) show the mixing ratio of ozone and the simulated accumulated ozone loss at the potential temperature level of 380 K.

Besides the simulated composition of chemical trace gases, CLaMS also allows displaying special model parameters. These are the so-called surface-origin tracers (Vogel et al., 2015) as well as the age spectrum (Ploeger and Birner, 2016). These quantities cannot be measured, but give insight of the history of the airmasses to be observed. Surface origin tracers are model

quantities that are initialized on the lower model boundary with 1 for a specific source area and with 0 elsewhere. These
quantities undergo transport and mixing and have no sink within the atmosphere. Thus the calculated numbers between 0 and
1 correspond to the fraction of air originating from the specific source area. The age of air reflects the time the specific air mass
has spent since it left a source area like the tropical troposphere. It is a typical measure of how long an air mass has spent in the
Brewer-Dobson circulation (Brewer, 1949; Dobson et al., 1946). Besides the mean age of air, also the so-called age-spectrum is
an important quantity. Due to mixing in the atmosphere, air masses with different history are always mixed. The age spectrum
is the calculation of the contribution to one specific airmass from the different origins (Ploeger and Birner, 2016). From that,
the following quantities have been derived for being displayed by MSS: (1) the fraction below 6 months of age of air, (2) the
fraction above 24 month of age of air and (3) the median of the age of air spectrum.

As an example for a linear view plot, Fig. 9 shows two trace gas species commonly used as stratospheric (ozone) and
tropospheric tracers (CFC-11).

### 3.1.5 Cirrus forecast

One further important research topic of SouthTRAC was the characterisation of cirrus clouds. Cirrus clouds are composed of
ice particles that nucleate upon cooling of the air by different processes. Depending on the number concentration and ice water
content, cirrus clouds either have a net cooling or net heating climate effect. The model package CLaMS-ice (e.g. Baumgartner
et al., 2022) was developed to simulate the characteristics of these ice particles using a two-moment bulk scheme (Spichtinger
and Gierens, 2009), that simulates ice water content (IWC) and ice particle number concentration. From this, also a mean
mass radius is calculated. This model is calculated typically along 24 h CLaMS backward trajectories while the water content
is initialized also from the ECMWF operational analysis or forecast. As this simulation is rather costly and scales with the
number of forecast points, these forecasts are typically provided in a limited resolution for only a few vertical levels.

### 3.2 Planning of a research flight

One of the main objectives of the SouthTRAC campaign was to measure the propagation of mountain waves into the polar
vortex (Rapp et al., 2021). Mountain waves are excited by the wind blowing across a mountain ridge. Forecasting mesoscale
phenomena like mountain waves calls for both accurate and accessible forecasts.

Mountain waves favour strong wind conditions and require an increase in wind speed with altitude for effective propagation
(Sato et al., 2012). A decreasing wind speed with height frequently induces wave dissipation (Geldenhuys et al., 2022). South-
TRAC was dominated by an anomalously early breakdown of the polar vortex (Fig. 7b). The evolving sudden stratospheric
warming triggered a rapid slow down in the westerly winds extending down from the polar vortex. This implied reduced
propagation heights for mountain waves.

The rapidly weakening polar vortex endangered the campaign objectives. Hence, the first scientific flight of the campaign
was aimed at measuring mountain waves from the Andes propagating as high as possible into the polar vortex. Thus knowledge
of the location of the polar vortex was of the utmost importance. During flight planning multiple figures like Fig. 7 and 10 were
consulted to ensure that the strong wind region (mountain wave pathway) would be probed by the instruments. The plan was to

cross the Andes mountains and align the flight track to the polar vortex winds at 1 hpa (points 1 to 7 on Fig. 10). By aligning it to the polar vortex winds, the propagation into the jet could be studied later.

The racetrack pattern (two parallel tracks — points 9 to 12 in Fig. 5) was designed for a dual purpose: the thermal limb sounder Gimballed Limb Observer for Radiance Imaging of the Atmosphere (GLORIA; e.g. Riese et al., 2014) was to observe the mountain waves below flight altitude (down to ≈5 km) and the Airborne Lidar for Studying the Middle Atmosphere (AL-IMA; Rapp et al., 2021) above (up to ≈80 km). The objective was to trace the same gravity wave packet from the troposphere to the lower mesosphere. In addition, the racetrack was meant to compensate for a weakness of the ALIMA instrument: it measures at high-resolution over a massive altitude range, but only in a 2-D vertical profile. A vertical profile provides a vertical wavelength and an along-track horizontal wavelength. However, by combining two racetracks, the phase orientation of the mountain wave can be estimated and thus an accurate horizontal wavelength computed (as done in Geldenhuys et al. (2022)).

Successful experiment design requires intimate knowledge of the instruments strengths and weaknesses. For example, the GLORIA instrument is a limb infrared imager with multiple strengths (e.g. allowing 3-D tomography), but one weakness is that water vapour in clouds saturates the signal and no measurement is obtained behind the cloud. For this reason cloud forecasts are of the utmost importance and are closely monitored during flight planning. By studying the vertical cross-section (Fig. 5c), MSUI allows one to easily adjust way points and flight days to minimise the impact of cloud cover. Another useful feature of the vertical cross-sections is to ensure desired targets are indeed measured by all instruments. The forecast in Fig. 5c shows mountain waves predominantly above the flight altitude with only some seen below over the main Andes (points 5 to 6) and in the racetrack over the south part of Patagonia (points 9 to 12).

Before the flight path is finalised, MSS products are used to check that the flight path do not violate, e.g., penguin protection zones, or commercial flight corridors. MSS is also used to check that the flight path do not exceed the range of the aircraft. If the range is exceeded, the flight waypoints are easily adjusted in MSS to reduce the range but to keep the scientific objectives intact. The flight plan was executed successfully on the 12th September 2019 and the obtained observations allow for the first gravity wave refraction study using high resolution observations (see Geldenhuys et al., 2022, for details).

## 4 Summary

The MSS software package is designed to support scientific aircraft measurement campaigns. It was put to regular and increasing use and greatly matured during the last decade. Thus, it has contributed to the successful implementation of the ML-Cirrus, NAWDEX, POLSTRACC, StratoClim, WISE, CoMet 1.0, EMeRGe, SouthTRAC, CIRRUS-HL, HALO-(AC)[3] and TI3GER aircraft measurement campaigns. The focus of development has been to improve the ease-of-use both for the actual flight planning, but also for setting up the MSWMS server component. Most recently, the addition of the MSColab component for collaborative flight planning has been found to be very useful during the Corona pandemic; beyond this, it also offers a central location for versioning planned and executed operations.

In particular, we used the MSS for flight planning of the SouthTRAC campaign. The data products we made available have proven very useful during the campaign. As an example, we described the reasoning behind the planning of the first scientific

flight and how the available data products supported the definition of the final way points. This demonstrates how the scientists can make the best use of MSS. More detailed documentation is beyond the scope of this paper, but is available on the web. MSS is an open source project that invites contributions by the community and so far, each campaign has contributed to its improvement.

*Code availability.* MSS has been published under the Apache License 2.0 Open Source Software license (Apache License (Version 2.0);
Mission Support System (MSS) on github). The most recent version is available on GitHub at https://github.com/Open-MSS. The current version v7.0.4 is also available at Zenodo at https://doi.org/10.5281/zenodo.7056451 (Bauer et al., 2022).

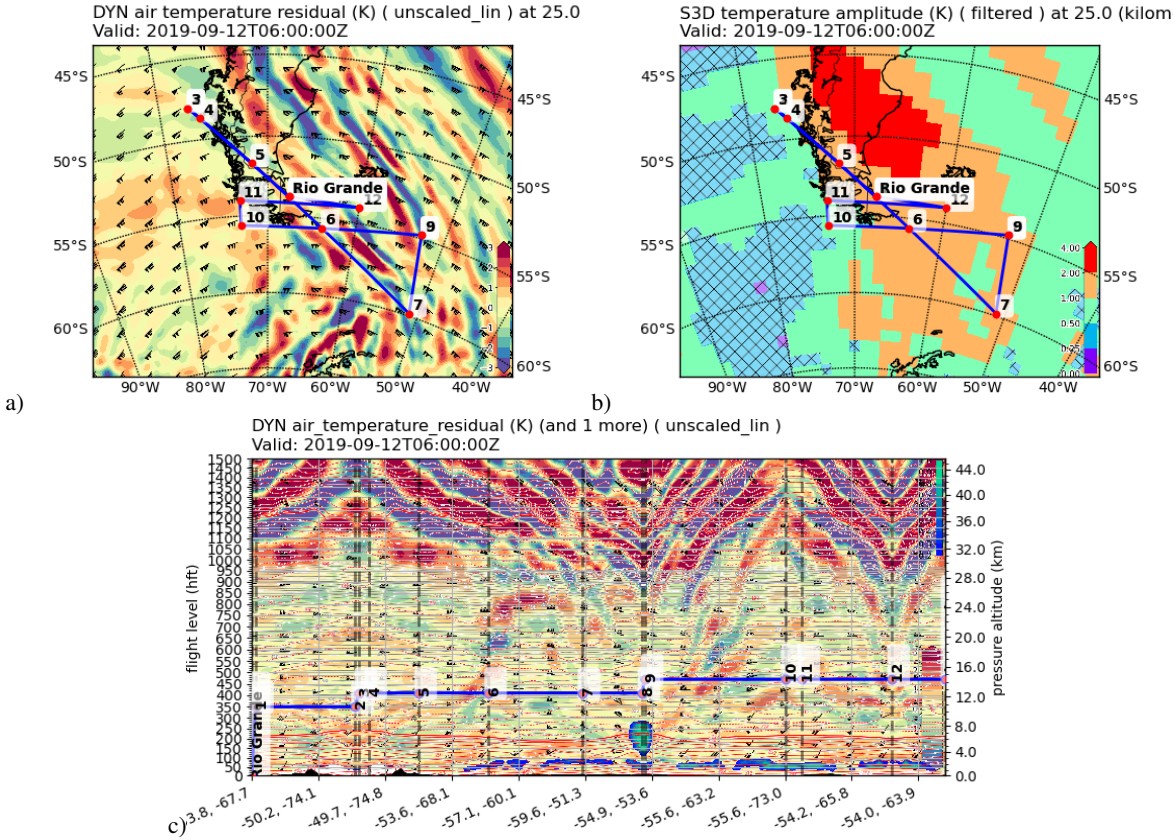

**Figure 5.** Gravity wave analysis figures: Temperature perturbations derived from ECMWF data on 25 km altitude (Panel **(a)**). Derived gravity wave amplitudes with hatched regions indicating high uncertainty at same altitude (Panel **(b)**). Temperature perturbations derived from one time step of ECMWF data sampled vertically along the planned flight path (Panel **(c)**); a cloud layer is displayed on top using the new multi-layering feature.

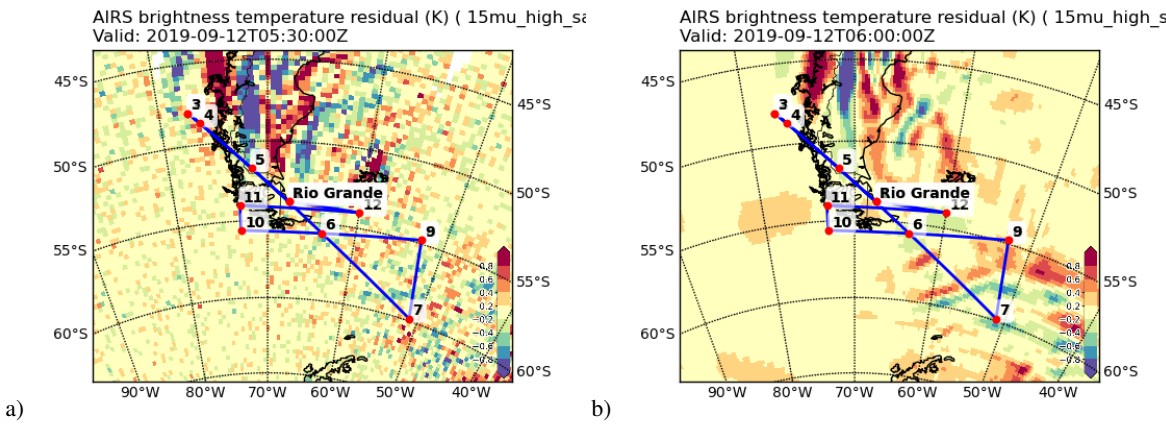

**Figure 6.** AIRS related figures: Brightness temperature perturbation from AIRS 15 $\mu$m measurements (Panel (**a**)). Brightness temperature perturbations predicted by ECMWF model temperatures (Panel (**b**)).

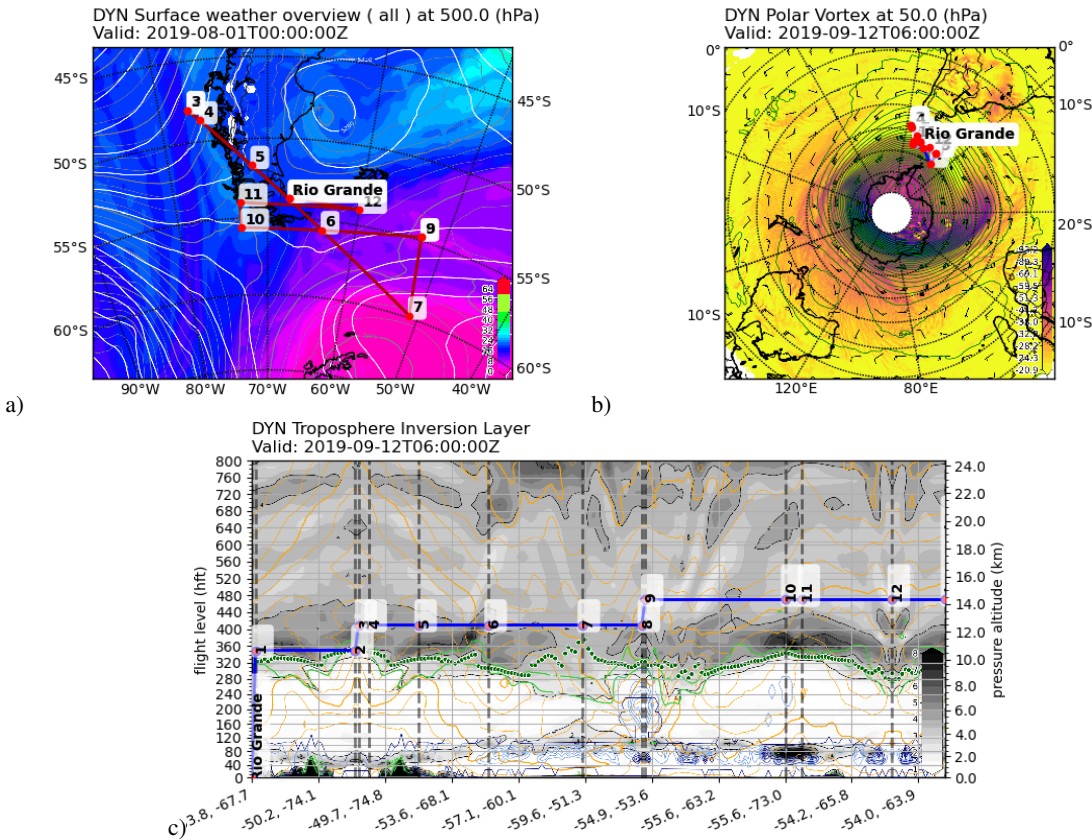

**Figure 7.** Dynamics related plots: Surface weather related plot (Panel **(a)**; equivalent potential temperature at 850 hPa (degC) as coloured contour surfaces, sea level pressure (hPa, grey), and geopotential height at 500 hPa (m, white) as contours). State of the polar vortex (Panel **(b)**; ertel potential vorticity at given altitude as coloured contour surface (PVU), geopotential altitude as green contours and wind speed as barbs). Static stability and tropopause inversion layer diagnostics (Panel **(c)**; square of Brunt-Vaisala frequency (grays, $1e4\,s^{-2}$), horizontal wind speed (orange, m/s), potential vorticity (green, PVU), cloud liquid water (navy, g/kg), cloud ice water (light blue, g/kg), thermal tropopause (dark green))

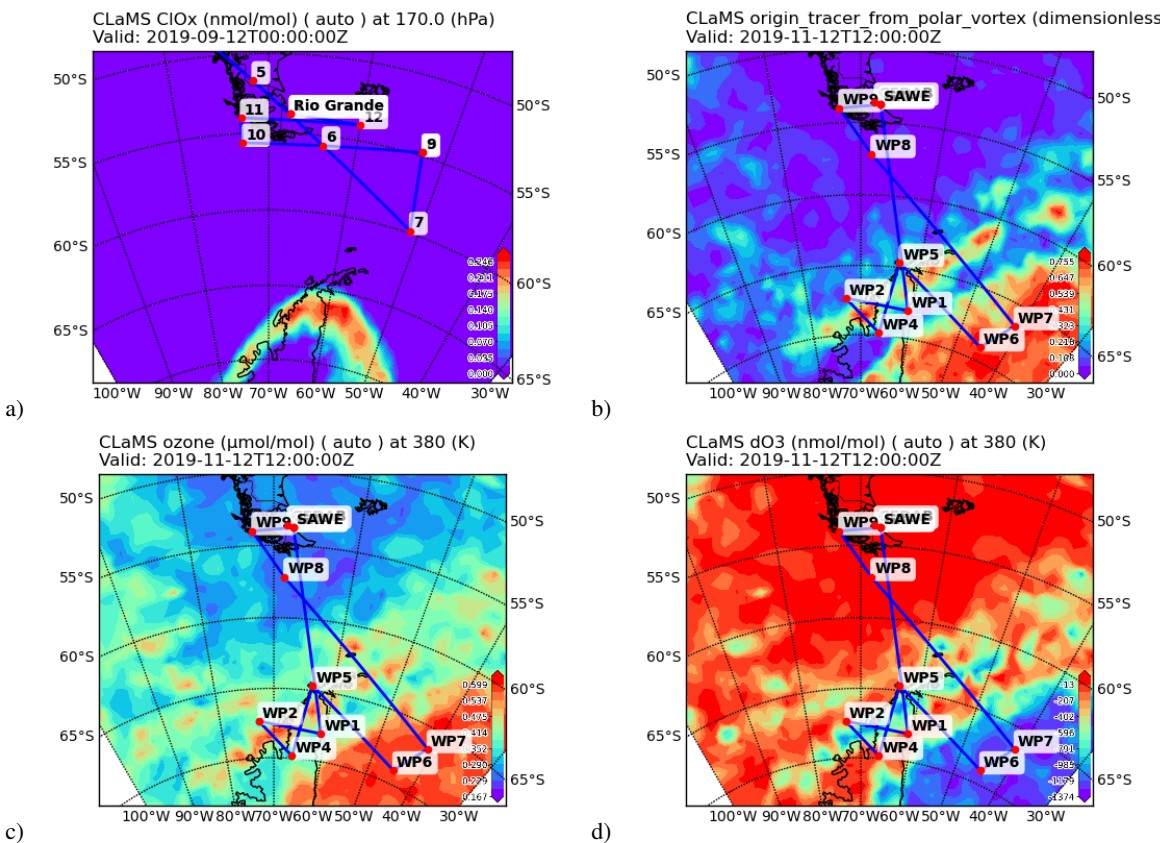

**Figure 8.** Chemistry related plots: active chlorine ClOx for flight ST08 (Panel **a**), the origin tracer from polar vortex for flight ST25 (Panel **b**), ozone mixing ratio (Panel **c**) and simulated accumulated ozone depletion (dO₃) (Panel **d**). Panel **a** is in cylindric projection while the others are in stereographic projection. Panels **c** and **d** also show the borders of the Flight Information Regions (FIR) as green lines (kml overlay).

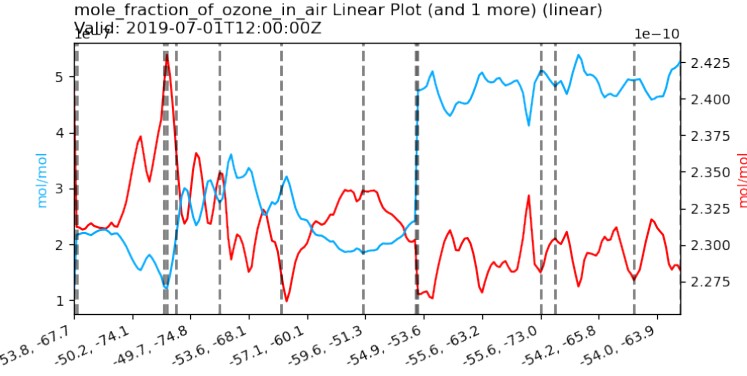

**Figure 9.** This shows an example of the linear view with ozone volume mixing rations (blue) and CFC-11 (red) sampled along the flight track.

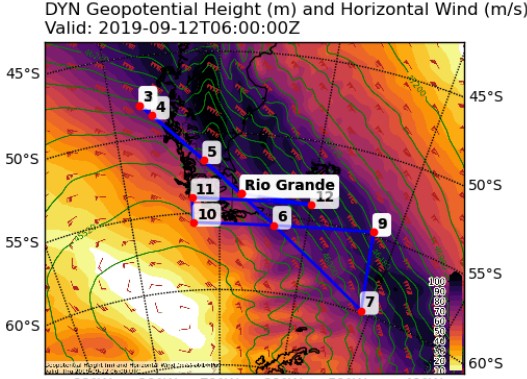

**Figure 10.** Horizontal wind speeds at 1 hPa. Geopotential altitude (m) in green. Wind direction and magnitude also barb.

## Appendix A: Technical features

MSS is implemented in the Python programming language with minor parts being written in JavaScript or Shell Script. The package is maintained as a Conda-Forge (Conda-forge community, 2015) package for easy installation (Mission Support System (MSS) on conda-forge). This allows the installation of MSS on the three major platforms Windows, MacOS, and Linux. MSS depends on many open source libraries such as NumPy, SciPy, MetPy, matplotlib, basemap, palletsprojects, git, owslib, PyQt5, pyfilesystem2, and socketIO.

The software is developed openly in the Open-MSS github project (Mission Support System (MSS) on github). There, it is continuously tested by a comprehensive test bench. We also maintain docker images for the purpose of testing or using any component of the software. Since 2019, several students have contributed significantly to the development of the project within the framework of Google Summer of Code (Google Summer of Code). By now (14th Sept 2022) the software had 3396 commits by 42 contributors representing 85062 lines of code (Mission Support System (MSS) on openhub).

The software is licensed according to the Apache License (Version 2.0).

The following sections describe some technical details interesting for users.

### A1  Installing MSS

The software is available on github and as an anaconda package on conda-forge. In addition, we provide automated installation scripts for Windows, Linux, and MacOS. Specific instructions can be found at https://mss.rtfd.io/en/stable/installation.html.

### A2  Import/Export Plugins

We found that different flight crews and scientists have widely varying requirements on which kind of data formats the flight tracks need to be provided in. Thus, MSS offers a simple way of adding plugins for importing and exporting flight track data in different formats. Included in the installation are several import/export filters located in the mslib.plugins.io module,

which also serve as an example for the definition of own plugins. With the Navigational Aids (NAVAID) Plugin for exporting flight path data, we give an example that writes out the waypoints in the NAVAID-DME (distance measuring equipment) formalism, that is defined by a radial distance and direction to the nearest NAVAID radio beacon (for documentation, see

https://mss.rtfd.io/en/stable/plugins.html).

## A3    Installing server components

All components are described in https://mss.rtfd.io/en/stable/components.html. For development and testing, we use simple and insecure built-in web-servers, but for production one should setup a more advanced robust server. For the MSWMS Server one can use any WSGI deployment. We provide currently in our documentation examples for the Apache Webserver. For the

MSColab server we use gunicorn and a nginx proxy.

## A4    Plot Gallery

For the MSWMS server definition all available plots can be created by the gallery option. An example based on a simple test data set is available within our documentation (https://mss.rtfd.io/en/stable/gallery/index.html). The same data can also be seen in Fig. A1.

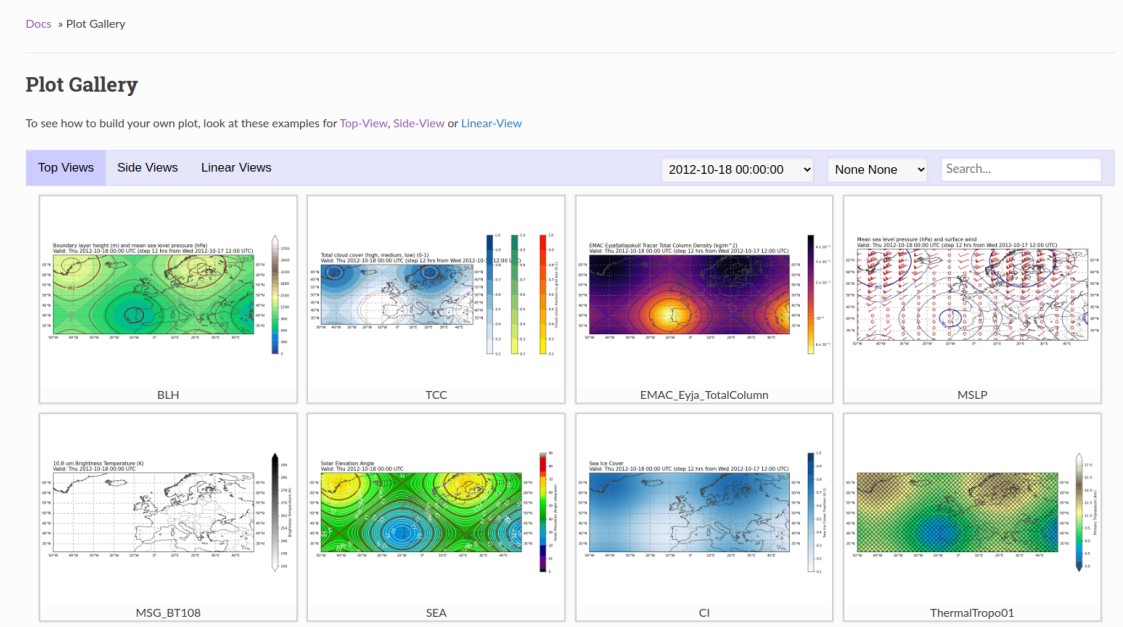

**Figure A1.** Example of plot gallery from the MSS Documentation

*Author contributions.* RB and JU developed or contributed to all work in this paper and wrote most of it. JUG did contribute the CLaMS related sections. LH provided the AIRS near-real-time gravity wave data products for integration into the MSS. MB did contribute largely to the MSS development of features described here. MG provided the section on flight planning for SouthTRAC. All authors reviewed the paper and provided textual improvements.

*Acknowledgements.* Atmospheric research with HALO is supported by the Priority Programme SPP 1294 of the Deutsche Forschungsgemeinschaft (DFG). This work was funded by the Deutsche Forschungsgemeinschaft (DFG, German Research Foundation) - UN 311/3-1. The European Centre for Medium-Range Weather Forecasts (ECMWF) is acknowledged for meteorological data support.

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
