# Peer review of "The Mission Support System (MSS v7.0.4) and its use in planning for the SouthTRAC aircraft campaign"

_Geoscientific Model Development, 2022_

## Author Comment (AC1)

We thank the reviewers for their review, interesting questions, and technical corrections.

All technical suggestions (wordings, abbreviations, etc.) have been applied if not indicated otherwise.

**1 Reply to Referee #1**

**1.1 Minor Comments**

1. *P2, l32: The tool has also been used during the ML-Cirrus campaign in 2014, the NAWDEX campaign in 2016, the CoMet 1.0 campaign in 2018 as well as the AC3 campaign in 2022.*

   We added these missing major campaigns.

2. *P3, Figure 1: introduce WSGI (Python Web Server Gateway Interface) in the main body of the text for completeness. What exactly is the "Data Tool Chain" indicated in the flow chart?*

   We moved the description of our example tool chain to the front and provided a consistent name for it (was called a "set of scripts" in the main text before).

3. *P9, l237: Knobloch et al. 2022 has been rejected*

   ...That is sad to hear. We changed the reference.

4. *P10: The chapter "conclusion" is rather a summary of the paper than a conclusion*

   We adapted the heading.

5. *The size of the figures should be increased as tick labels and figure titles are hard to read. I guess it's not the fault of the authors, but probably more a result of using the manuscript template. Most figures seem to be vector graphics, anyhow. Hence, it shouldn't be a problem to increase their size for a possible publication of the manuscript.*

   Most figures are generated, necessarily, by the MSS software and are thus designed to be viewed in fullscreen mode on a display. However, we regenerated all MSS-figures using a different desktop configuration with larger fonts and boundary boxes to facilitate viewing of printed figures in the Journal.

**2 Reply to Referee #2**

**2.1 General comments**

1. *Font sizes of all figures are very small. In particular screen shots of MSUI windows or plots ar so small that it is hard to read any of the text on buttons or axes. Sometimes I need to zoom up to 300% into the figure to read anything.*

   The figures are generated by the MSS software and are thus designed to be viewed in fullscreen mode. However, we regenerated all figures using a different desktop configuration and boundary boxes to facilitate viewing in the Journal.

2. *At some points of the manuscript, it seems like this paper is aiming to be a documentation for MSS (e.g. when the names of certain python scripts are mentioned for certain settings). This may be helpful for MSS users, but is not very consistent throughout the paper. I would suggest to leave such details for the documentation and mention this documentation instead.*

   The detail level w.r.t. to the description of the program was indeed varying during the development of the paper and brought to a more generic level. We removed some detailed

references and focus on providing the "big picture". The referenced documentation delivers, indeed, all the technical details for any interested party.

**2.2 Specific comments**

1. *Line 61: "Windy (win)" - is "win" the abbreviation of Windy?*

   The cite of the website by the 'Windy' company malfunctioned. We fixed this.

2. *Figures 1 and 2: The text in this figure is hard to read without zooming in a lot.*

   We rearranged the boxes and increased the font sizes.

3. *Line 89: "This configures the desired plotting layers": What is exactly meant by "plotting layers"? Are these the options of horizontal cuts through the atmosphere that are available for plotting?*

   This is correct. We explain the term after its first use. The plotting layers are python classes, which can also be defined and configured in the user-defined mswms_settings.py file, which define the available maps and cross-section types.

4. *Line 105 and following: It would be nice to have examples of the side view and linear view plot. In later figures, side view plots are shown, so it would be good to mention these figures as examples here. Linear view plots would be also nice to see as an example.*

   We expanded the text with an example of the linear view in the CLaMS model related section.

5. *Line 115: Same for the table view: An example figure would be great!*

   We added a table view figure here, which allows to discuss the features in more detail.

6. *Introduction of section 3: The campaign took place already almost three years ago. Are there some exemplary papers published for the aimed objectives apart from the overview paper by Rapp et al.?*

   We expanded the section and listed some of the most important results.

7. *Figure 4c: It seems like this flight is planned to take several hours. Does MSS account for the flight time and potential changes in the forecast during that time? Or is this side view rather a snapshot for a single model time?*

   At the current point in time, MSS can only provide snapshots of provided model information. Typically one steps through the time steps available for the flight duration to get an impression of the changes over time and, via the table view, can align the waypoints with time and adopt the plan. We'd like to add a feature that would perform an interpolation of model data on the flightpath not only in space, but also in time (see https://github.com/Open-MSS/MSS/issues/803). The authors do not currently have the ressources for doing so, but hope to find time or contributors in the near future.

   We adopted the caption to make this clear.

8. *Line 175: "The forecast data product has proven to be largely reliable for forecasting gravity wave structures visible to AIRS.": What "forecast data product" is meant in this sentence?*

   We expanded the text to explain that we refer to the brightness temperature residual data product.

9. *Figure 5: The title of panels a and b are almost the same except of the "valid" time. But it seems obvious that the figure caption is right here that panel a shows a measurement and panel b shows a simulation. So, I guess that the title of panel b is wrong, since it does not show AIRS data.*

The server generating the figure was not configured in a way that the title between the two data sets would be different. This has been corrected now and the figure has been updated.

10. *Section 3.1.3 does only contain two sentences and could be either expanded or omitted.*

Following the suggestion of the reviewer, we expanded on the section.

11. *Section 3.2: It is great that examples are mentioned in such detail and that way points are given for certain figures. But unfortunately, it is very difficult to recognize these way points without zooming into the figure up to 300%.*

The figures are generated by the MSS software and are thus designed to be viewed in fullscreen mode. However, we regenerated all figures using a different desktop configuration and boundary boxes to facilitate viewing in the Journal.